# Heavy Metal Bioaccumulation in Peruvian Food and Medicinal Products

**DOI:** 10.3390/foods13050762

**Published:** 2024-02-29

**Authors:** Teresa R. Tejada-Purizaca, Pablo A. Garcia-Chevesich, Juana Ticona-Quea, Gisella Martínez, Kattia Martínez, Lino Morales-Paredes, Giuliana Romero-Mariscal, Armando Arenazas-Rodríguez, Gary Vanzin, Jonathan O. Sharp, John E. McCray

**Affiliations:** 1Facultad de Ingeniería de Procesos, Universidad Nacional de San Agustín de Arequipa, Arequipa 04001, Peru; ttejadap@unsa.edu.pe; 2Department of Civil and Environmental Engineering, Colorado School of Mines, Golden, CO 80401, USAjsharp@mines.edu (J.O.S.); jmccray@mines.edu (J.E.M.); 3Intergubernamental Hydrological Programme, United Nations Educational, Scientific, and Cultural Organization (UNESCO), Montevideo 11200, Uruguay; 4Departamento Académico de Química, Facultad de Ciencias Naturales y Formales, Universidad Nacional de San Agustín de Arequipa, Arequipa 04001, Peru; jticonaq@unsa.edu.pe (J.T.-Q.); lmoralesp@unsa.edu.pe (L.M.-P.); 5Facultad de Geología, Geofísica y Minas, Universidad Nacional de San Agustín de Arequipa, Arequipa 04001, Peru; omartinez@unsa.edu.pe; 6Facultad de Ciencias Naturales y Formales, Universidad Nacional de San Agustín de Arequipa, Arequipa 04001, Peru; kmartinezr@unsa.edu.pe; 7Escuela de Ingeniería Ambiental, Facultad de Ingeniería de Procesos, Universidad Nacional de San Agustín de Arequipa, Arequipa 04001, Peru; gromeroma@unsa.edu.pe; 8Departamento Académico de Biología, Facultad de Ciencias Biológicas, Universidad Nacional de San Agustín de Arequipa, Arequipa 04001, Peru; aarenazas@unsa.edu.pe; 9Hydrologic Science and Engineering Program, Colorado School of Mines, Golden, CO 80401, USA

**Keywords:** plants, fish, mollusks, macroinvertebrates, mammals, codex, regulations

## Abstract

To better query regional sources of metal(loid) exposure in an under-communicated region, available scientific literature from 50 national universities (undergraduate and graduate theses and dissertations), peer-reviewed journals, and reports published in Spanish and English were synthesized with a focus on metal(loid) bioaccumulation in Peruvian food and medicinal products utilized locally. The study considered 16 metal(loid)s that are known to exert toxic impacts on humans (Hg, Al, Sb, As, Ba, Be, Cd, Cr, Sn, Ni, Ag, Pb, Se, Tl, Ti, and U). A total of 1907 individual analyses contained within 231 scientific publications largely conducted by Peruvian universities were analyzed. These analyses encompassed 239 reported species classified into five main food/medicinal groups—plants, fish, macroinvertebrates and mollusks, mammals, and “others” category. Our benchmark for comparison was the World Health Organization (Codex Alimentarius) standards. The organisms most frequently investigated included plants such as asparagus, corn, cacao, and rice; fish varieties like trout, tuna, and catfish; macroinvertebrates and mollusks including crab and shrimp; mammals such as alpaca, cow, chicken eggs, and milk; and other categories represented by propolis, honey, lichen, and edible frog. Bioaccumulation-related research increased from 2 to more than 25 publications per year between 2006 and 2022. The results indicate that Peruvian food and natural medicinal products can have dangerous levels of metal(loid)s, which can cause health problems for consumers. Many common and uncommon food/medicinal products and harmful metals identified in this analysis are not regulated on the WHO’s advisory lists, suggesting the urgent need for stronger regulations to ensure public safety. In general, Cd and Pb are the metals that violated WHO standards the most, although commonly non-WHO regulated metals such as Hg, Al, As, Cr, and Ni are also a concern. Metal concentrations found in Peru are on many occasions much higher than what has been reported elsewhere. We conclude that determining the safety of food/medicinal products is challenging due to varying metal concentrations that are influenced not only by metal type but also geographical location. Given the scarcity of research findings in many regions of Peru, urgent attention is required to address this critical knowledge gap and implement effective regulatory measures to protect public health.

## 1. Introduction

Being located next to the South American Andes, Peru is rich in water resources and an important food producer. The country is ranked among the main global exporters of grapes, avocados, mangos, asparagus, blackberries, bananas, ginger, and onions, among many other food products, with transactions of more than USD 13.8 billion/year [1]. However, in March 2022, the EU banned Peruvian avocados due to the high contents (over 0.05 mg/kg, the legal limit) of Cd [2]. In the same year, more Peruvian avocados (e.g., 0.03, 0.014, and 0.004 mg/kg above MPL on different occasions), peppers (Cd 0.07 mg/kg above MPL), Angola peas (Cd 0.013 mg/kg above MPL), organic cacao powder (Cd 0.6 mg/kg above MPL), chocolate (Pb 2.7 mg/kg above MPL), and organic ginger root (0.06 mg/kg above MPL), among other food products, were also banned for having high heavy metal contents [3]. Not surprisingly, Peruvian investigators have quantified heavy metals in many food products; for example, Chirinos-Peinado et al. [4] reported mean Cd concentrations greater than 500 mg/L in 40 milk samples collected near the Mantaro River (central Peru), which overpassed the legal limit nearly 200 times, through biological uptake and bioaccumulation [5]. Other Peruvian studies have reported metal content in food products such as cow (*Bos taurus*) milk in Moquegua (0.29 mg/kg of As) [6] and rainbow trout (*Oncorhynchus mykiss*) in Junín (6.11 mg/kg of Pb) [7], while Gutleb et al. [8] detected Hg levels above MPL in catfish (*Pimelodus ornatus*) in Madre de Dios (0.61 mg/kg).

Excessive levels of metal(loid)s (referred to as metals) in food products have a direct and significant impact on public health (e.g., [9]). A recent comprehensive review conducted by Skalny et al. [10] highlighted the culmination of two decades of research, revealing an association between Hg exposure and cancer. Additionally, various studies have linked Hg to psychological alterations, pregnancy issues, and other health concerns [11,12]. Another review by Suhair et al. [13] emphasized the adverse effects of Pb ingestion on children’s health, including conditions such as asthma and learning disabilities. Further investigations have indicated that Pb can also contribute to neurobehavioral issues, hearing and muscle functioning problems [14], permanent intellectual damage [15], and cardiovascular problems [16], among others (e.g., [17]). Cr, Cd, and As have been identified as substances capable of causing organ damage and various severe medical conditions [18,19,20]. Moreover, numerous other metals (detailed in Table 1) have been associated with a range of health consequences in humans (e.g., [21]).

As national and export food demands increase in Peru (e.g., [22]), the country faces serious public health and, by extension, economic challenges associated with the presence and bioaccumulation of metals in food and medicinal products. The “Platform of People Affected by Toxic Metals”, a Peruvian organization that seeks to protect victims of this growing environmental problem, estimates that more than 10 million Peruvian citizens are somehow affected by metals ingested through food and medicinal products. Taking this into account, the number of Peruvian studies that focus on this important topic (metal bioaccumulation in food and medicinal products) and published in peer-reviewed English journals has increased significantly in recent years (see Figure 1 and details in Appendix A), where crops, fish and medicinal plants are the primary research topics.

Although human exposure to toxic amounts of certain metals, such as arsenic, has existed since pre-industrial times in the region [23], recent studies [24,25] suggest that mining operations are increasing this exposure in the Peruvian Andes. In addition to impacts on commercialization and export [26], this exerts ecotoxicological effects as discussed by Herrera-Perez and Mendez [27], Custodio et al. [28,29], Molloy et al. [30], Weinhouse et al. [25], and Gerson et al. [31]. Despite a trending increase in research, as documented in Figure 1 and Appendix A (a total of 52 Peruvian studies related to bioaccumulation published in the English scientific literature), these studies have not incorporated a wealth of local knowledge and findings published in Spanish, providing motivation to conduct a national analysis that compiles and synthesizes this knowledge.

**Table 1 foods-13-00762-t001:** Origins and health effects of metals included in this review.

Metal	Health Effects	Main Source	References
Hg	Metallic taste, bad breath, nausea, vomiting, diarrhea and sporadically brownish discoloration of the incisors, emotional sensitivity, dementia, kidney damage, dermatitis, conjunctivitis, damage to color vision, and rhinitis	Mining activity (gold mining)	[32]
Al	Respiratory, neurological, and neurodegenerative diseases such as Parkinson’s	The mining industry, in mineral processing, in the production of alloys or metallic aluminum, and power plants that use coal	[33,34]
Sb	Hypertension, diabetes, liver damage, decayed teeth, carcinogenic effects, and polycystic ovary (breast cancer, prostate cancer, and colorectal cancer)	Natural, industrial activities (painting and batteries), and mining.	[35,36]
As	Skin lesions, cancer (different types), and arsenicosis	In naturally occurring rocks and minerals, wood that has been treated with copper chromate arsenate, and some pesticides	[33,37,38,39]
Ba	Acute gastroenteritis and loss of reflexes (muscle paralysis).	Natural and industrial activities (paints, cement, and fertilizers)	[40]
Be	Acute or chronic allergic or irritative reactions, berylliosis, and cancer	Naturally in rocks, coal, sediments, volcanic material, which can thus contaminate groundwater	[33,41]
Cd	Anemia, kidney dysfunction, osteoporosis, osteomalacia, respiratory problems, hypertension, nervous disorders, weight and appetite loss, and prostate and lung cancer	Natural (minerals and volcanic activity) and industrial activities such as production of batteries, phosphate fertilizers, detergents and refined petroleum products, paints, plastics, and zinc refining	[42]
Sn	Fatigue, headaches, diarrhea, vomiting, muscle weakness, anemia, liver and kidney damage, and reduction in neurotransmitters in the brain	Rocks with natural tin content, tin mines, manufacturing industries that use tin such as in making plastics, food packaging, pesticides, and rodent repellents	[33,43]
Ni	Skin conditions; dizziness; asthma; chronic bronchitis; lung embolism; respiratory failure; birth defects; heart disorders; allergic reactions; and lung, nose, larynx and prostate cancer	Industrial activities (metallurgy, batteries, solar equipment, and galvanization)	[44]
Ag	Damage to the alveoli, argyria, and mild allergic skin reactions	In water and air naturally via the erosion of rocks or minerals, mineral processing, and photo processing	[33,45]
Pb	Kidney dysfunction, kidney cancer, encephalopathy, poor attention span, memory loss and hallucinations chronic damage to the Central Nervous System (CNS) and Peripheral Nervous System (PNS), coronary heart disease, and cardiovascular functional abnormalities	Mining, industrial activities (batteries, paints, gas stations, glass production, and plastics)	[46,47]
Se	Keshan disease, cancer, type 2 diabetes, and diseases of the endocrine system and circulatory system	Naturally in rocks and soils and from agricultural and industrial waste (paints, enamels, rubber, dyes, fungicides, etc.)	[33,48]
Tl	Gastrointestinal symptoms, hair loss, and permanent neurological damage, which can be fatal	Rodenticides and pesticides	[49]
Ti	Hepatic damage	Factories that produce titanium tetrachloride	[33,50]
U	Kidney failure, decreased bone growth, DNA damage, carcinogenesis, and mutations in animals	Natural (groundwater), nuclear power plants, and industrial activities (catalysts and pigments)	[51,52]

Galagarza et al. [53] developed a review of the chemical contaminants (including metals) contained in Peruvian plant-based food products. For the specific case of metals, the authors developed a review of six inorganic pollutants (Pb, Zn, Cu, Cd, As, and Cr) in 28 food products documented in 16 studies [54,55,56,57,58,59,60,61,62,63,64,65,66,67,68,69]. Building upon their findings, we conducted an extensive literature search of scientific publications related to metal bioaccumulation in Spanish (undergraduate and graduate theses from 50 public universities as well as peer reviewed journals and reports) considering 16 metals that are known to exert toxic impacts on humans (Hg, Al, Sb, As, Ba, Be, Cd, Cr, Sn, Ni, Ag, Pb, Se, Tl, Ti, and U because they were evaluated in the country’s investigations) and bioaccumulated in food and medicinal products common to the Peruvian diet. For each study and food type evaluated (including medicinal and topical products), the maximum documented metal values were considered. Outliers associated with metal-based laboratory extractions, plants used for bioremediation in post-mining and other polluted soils, or possible measuring errors were excluded from the statistical analysis, which consisted of creating boxplots to provide a general idea of the national distribution of metal concentrations, as well as calculating the national mean and standard deviation for each metal. Throughout the manuscript, where enough data existed for statistical comparisons, we evaluated metal concentration according to logical groupings such as species or material (tissue, body parts, etc.), using Tukey’s honestly significant difference (HSD) test within the statistical software JMP [70]. A *p*-value less than 0.05 was considered statistically relevant. As the reports varied with respect to normalization, metal concentrations were normalized to mg of metal per kg of product for consistency. Then, these were tabulated and classified into different food or medicinal categories, each with independent statistical analyses. The obtained values were compared with the MPL of the World Health Organization (WHO, Codex Alimentarius) for contaminants in food and medicinal products (Appendix A), identifying metals and food products that were below the norm, above it, or not regulated (i.e., either the metal or the species is not included in the WHO standards). This national evaluation included the geographical and temporal evolution of the studies (and individual analyses) performed in Peru as well as which research institutions are leading metal uptake and bioaccumulation studies in the country. Finally, for consistency and better organization, food and medicinal products were differentiated and combined in the same categories (i.e., when a product is edible and medicinal), giving preference first to food sources. This analysis provides the most comprehensive database to date of metal bioaccumulation in food/medicinal products consumed domestically in Peru. Therefore, it provides a foundation for a further understanding of the mechanisms of human exposure that could help inform regulatory actions that protect local inhabitants and could increase the economic viability of Peru’s export to the international market.

## 2. Findings and Discussion

### 2.1. Comparison to WHO Advisory Lists

A significant portion of the metal concentrations found in this national review were above WHO standards, where Cd and Pb are the most problematic metals (see Table 2). However, what drew the most attention is the fact that most of the documented metals in the country are currently not regulated under the WHO system in any of the food-type classifications, suggesting an urgent need to regulate the toxic contaminants that Peruvians are consuming through the food/medicinal products found in this investigation. Assuming that the diversity of food products is increasing and given that WHO benchmarks miss some prominent sources of nutrition consumed by Peruvian citizens, there is a need to refine and update recommendations to better protect public health in Peru. For example, the categories macroinvertebrates and mollusks and mammals (see Section 2.2 for details) have only one regulated metal (Cd and Pb, respectively) (Table 2), an indication that changes in the regulatory system are needed.

Based on a simple analysis of several current international food regulation systems, there is a large inconsistency in MPL values among the standards as well as the metals involved. Table 3 shows a general comparison of metals involved in different international food standard systems. Most standards consider Total Hg (and/or, methylmercury, CH_3_Hg), As (Total and/or Inorganic), Cd, Sn, and Pb, with only one including Sb (South African). While Hg is regulated in its elemental form, some standards specifically request MPLs for CH_3_Hg (the WHO, Cuba, and S. Africa). Similarly, As is most times required as Total (As(1) in Table 3), but it is also sometimes required in its inorganic form, or As(2) in Table 3 (Australia–NZ, S. Africa). Additionally, MERCOSUR, an acronym for “Mercado Común del Sur” (Southern Common Market), was established as a regional trade agreement in 1991 with four founding nation states: Argentina, Brazil, Uruguay, and Paraguay, and it is one of the standards with the most food categories. Taking into account the wide range of metals found in Peruvian food/medicinal products through this review, national standards should be based not only on the WHO system (which the country currently is under), but also on South African standards, including an additional number of toxic metals found in this analysis as well (Al, Ba, Be, Ni, Ag, Se, Tl, and Ti), all designed so that each product that any Peruvian consumes should not contain metals above toxic concentrations.

Worldwide, the most commonly controlled metal is Pb (280 food categories) followed by Cd, As (Total), Hg, and Sn. For the specific case of the WHO (see Appendix A), the most widely used advisory body in the world (including Peru), As is present in six food categories including water and salt, two categories for rice, and the remaining categories of edible fats and oils, and fat spreads and blended spreads. Similarly, 17 categories include Cd, which are water and salt, two types of chocolate, marine bivalved mollusks, and cephalopods, and the rest are plant-related food products. Pb, on the other hand, is considered in 41 categories, including water and salt, canned fruits, jams, jellies and marmalades, mushrooms, canned vegetables, fruit juices, infant formula, fish and meat (cattle, pig, sheep, and poultry), milk and milk products, wine and liqueur, and specific products such as canned chestnuts and canned chestnut puree. Hg (a very common contaminant found in this review) is considered just for water and salt, while CH_3_-Hg is considered according to the WHO standards in only four species of fish (tuna, alfonsino, marlin, and shark), excluding fish species found in this review; the WHO norm states: “If the total mercury concentration is below or equal to the MPL for methylmercury, no further testing is required, and the sample is determined to be compliant with the MPL. If the total mercury concentration is above the ML for methylmercury, follow-up testing shall be conducted to determine if the methylmercury concentration is above the MPL”. Finally, Sn is included in seven WHO categories (canned food, canned brewages, and specific types of canned meat and ham). However, despite their tremendous global influence, the WHO standards generally do not include important metals (and food/medicinal products) that are present in Peru (see Table 2). Moreover, metals such as Ba, Ag, Tl, and Ti, which have been identified in this review (Section 2.3), are not included in any international standards for MPLs despite their toxicity and potential harmfulness to humans. For example, Ba was found in this review and is considered toxic [71], being reported in various food products around the world (e.g., [72,73,74]). According to a report released by the US Department of Health and Human Services (Agency for Toxic Substances and Disease Registry) [75], excessive ingestion of Ba can cause paralysis; the largest Ba concentration in Peru was documented by Carpio [76] for the medicinal plant Yana llachu (*Elodea potamogeton*) (2.77 mg/kg) and by Huancaré [77] in rainbow trout’s muscular tissue (15.57 mg/kg), both being widely consumed by Peruvians. Similarly, though evaluated on nine occasions and not showing any presence up to date other than in rainbow trout’s kidney by Acosta [78] (0.003 mg/kg), U has been linked to kidney failure and death and has been documented to be released from atrophic activities such as mining (e.g., [79]), so its regulation might not be justified for Peru. All the above represent just examples of how important it is to monitor toxic metals that are present in actual food/medicinal products that are normally consumed in Peru.

In addition to the above, there are clear inconsistencies among international standard systems in terms of metals and food products considered in this Peruvian review. For example, Table 4 details the metal MPLs included in seven international standards under the general categories “Fish” and “Fruits”, both showing how heterogeneous the international system currently is, with the Cuban standards being the most complete (for these two particular cases). Importantly, most countries/regions differ in terms of how each classifies food, which ranges from highly general to specific. A clear example is represented by the category “Fish” (Table 4), in which some countries/regions classify them just as “Fish” (Cuba, Australia–New Zealand, Brazil, South Africa, and Switzerland), while others have additional MPLs for some specific species (e.g., the WHO and EU). Similar situations occur for some vegetables, processed foods, and animal meat, among many others, suggesting the need to use international standards that can be tailored to specific countries and regions such as Peru.

### 2.2. Metal Bioaccumulation in Peruvian Food/Medicinal Products

A total of 231 Spanish-language scientific publications were used to generate a centralized database of metal concentrations in different Peruvian food/medicinal products (Table 5). Those publications were distributed among undergraduate university theses (a total of 143 publications, 62%), graduate theses and dissertations (55, 24%), peer-reviewed journals (22, 10%), and government and research reports (11, 5%). Peruvian national universities played a prominent role, with 7 of them having at least 10 publications; conversely, only 18 public universities (out of 50) did not show any research related to this topic (see Appendix A). Building upon the findings from this review, further bioaccumulation-related national reviews should include the private educational sector (92 additional universities nationwide) [80], which represents a valuable source of information that needs to be explored.

This broadly reaching data collection resulted in documentation of metal bioaccumulation across a total of 239 species. As summarized in Table 5, the species identified from this dataset were subsequently organized into five main categories: plants (Appendix A), fish (Appendix A), macroinvertebrates and mollusks (Appendix A), mammals (Appendix A), and other food types that did not clearly classify in the above categories (Appendix A).

Most publications (59%) investigated bioaccumulation in plants, followed by fish (21%) and macroinvertebrates and mollusks (10%), with mammals and other categories representing 5% each. Considering all studies included in this review, a total of 1907 individual analyses were found, with plants (1211 analyses, 64%) being the most evaluated food category, followed by fish (284 analyses, 15%), macroinvertebrates and mollusks (214 analyses, 11%), mammals (105 analyses, 6%), and other categories (93 analyses, 5%). The number of scientific publications related to bioaccumulation in Peru has increased with time, with a temporary potential dip in 2020 that coincided with the height of the COVID-19 pandemic (Figure 2). The findings from each above-mentioned food category classification are discussed in the following subsections.

The geographic distribution of scientific publications found in this national review is illustrated in Figure 3. Most publications related to metal bioaccumulation are concentrated in the Department of Lima (10%) followed by Junin and La Libertad (8% each), Arequipa (7%), and Ucayaly (7%). As discussed later, this geographic distribution can be linked to local resources. However, it is important to mention that Lima has ten national universities, while regions usually have between one and three, which could explain the large concentration of studies from the country’s capital department.

### 2.3. Metal Bioaccumulation According to Food Category

#### 2.3.1. Plants

Plant food products were by far the category with the most reported cases of metal bioaccumulation in Peru (1211 documented analyzes, as previously mentioned). In total, 154 plant species were evaluated for metal bioaccumulation in Peru. These plants were further categorized under leaves (201 analyses, 17%), underground components (143 analyses, 12%); fruits and grains (167 analyses, 14%); seeds and their parts (141 analyses, 12%); peel, stems, and bark (142 analyses, 12%); processed foods (197 analyses, 16%); mixtures (205 analyses, 17%); and other categories (15 analyses, 1%), indicating that in general, plant components have been well studied in terms of metal content, at a national level. Appendix A details the metal concentrations of all plants, plant portions, and plant-derived food/medicinal products, including products below (311 analyses, 26%) and above (197 analyses, 16%) the WHO standards as well as those currently not regulated (703 analyses, 58%) (see Section 2.1 for more details).

The statistical analysis revealed that aerial, non-reproductive tissues such as leaves, stems, peels, and bark bioaccumulate metals at higher concentrations than underground components, fruits, grains, and processed or mixtures of materials. For example, As is higher in tissues such as peels, stems, and bark than in fruits, grains, and processed material, while Cr is higher in leaves than in fruits and grains, mixtures, and processed material. Leaves also appear to bioaccumulate Hg in higher concentrations in leaves relative to fruits, grains, and processed components. Data for Ni concentration were available for the processed and mixtures category and were found to be higher in the processed category, attributable to Ni concentration in teabags (see Appendix A for details).

Among the most common plant-related food products containing metal concentrations above the WHO’s standards are those high in Cd such as medicinal chocho (*Lupinus mutabilis*) stems (50.4 mg/kg), roots (46.4 mg/kg), and leaves (16.5 mg/kg) [81]; roots from pastureweed (*Cyathula prostrata*) (11.2 mg/kg) [82]; and potato (*Solanum tuberosum*) leaves (11.1 mg/kg) [83]. Moreover, common edible plant-based products above Cd-related MPLs are asparagus (*Asparagus officinalis*) sprouts (4.4 mg/kg) [84] and corn (*Zea mays*) grains (3 mg/kg) [85]. Another clearly problematic regulated metal in Peruvian plant-based products is Pb, highlighting medicinal species such as queñua (*Polylepis* sp.) (228.6 mg/kg) [86], senecio (*Senecio rufescens*) (64.8 mg/kg) [87], and fig (*Ficus nitida*) (54.9 mg/kg) [88] leaves, as well as edible species corn (13.1 mg/kg) [85] and cacao (*Theobroma cacao*) fruit (9.9 mg/kg) [89]. Arsenic was also present above regulatory thresholds in rice (*Oryza sativa*) grains (0.6 mg/kg) [90], which is higher than prior documentation of the concentrations in Australian rice (0.14 mg/kg) as reported by Rahman et al. [91]. Despite the above, the most concerning fact is the clear presence of high metal concentrations and/or plant-based products that are not regulated by the WHO standards. For example, high metal concentrations were detected in medicinal products such as arctic rush (*Juncus arcticus*) (50.1 mg/kg of As) [92], remocaspi (*Aspidosperma rigidum*) bark (7.1 mg/kg of Hg) [93], and huacapurana (*Campsiandra angustifolia*) leaves (17.5 mg/kg of Cr) [94]. Similarly, among the edible plant-based products reported with highest non-regulated metal concentrations are cacao seeds (6.9 and 30.9 mg/kg of Cd and Pb, respectively) [95], camomille (*Matricaria chamomilla*) teabags (15.7 mg/kg of Pb) [96], glasswort (*Sarcocornia fruticosa*) stems (1.6 mg/kg of As) [97], suggarcain (*Saccharum officinarum*) juice (0.5 mg/kg of Hg) [98], and yuca (*Manihot esculenta*) tuber (4.1 mg/kg of Cr) [99], among others (see Appendix A for details). Unfortunately, the number of plant-based edible/medicinal products found in this review was so large that making a comparative analysis (including metal concentrations reported in other countries) is not practical. However, the information contained in Appendix A is valuable and should be a starting point for further investigations.

Additionally, a number of plant species contained higher metal concentrations. Arctic rush (*Junkus arcticus*) was statistically higher in As than 86 other species, while tarwi (*Lupinus mutabilis*) was higher than 18 species, and remocaspi (*Aspidosperma rigidum*) was higher than 5 species. Cadmium was higher in chocho (*Lupinus mutabilis*) when compared to 149 other plant species, while pastureweed (*Cyathula prostrata*) was higher than 109 species. For Cr, huacapurana (*Campsiandra angustifolia*) was statistically higher than 14 other species, while remocaspi was higher than 2. Mercury in remo caspi (*Aspidosperma excelsum*) was higher than 59 other species, while hinojo llachu (*Mysiophyllum elatinoides*) and yana llachu (*Elodea potamogeton*) were higher than 8 and 4 species, respectively. Six species were elevated in Pb concentrations relative to other species in the data set, including figs (*Ficus nitida*), panamito bean (*Phaseolus Vulgaris*), queñua (*Polylepis* sp.), senecio (*Senecio rufescens*), manayupa (*Desmodium molliculum*), and anis (*Pimpinella anisum*).

Based on Appendix A, medicinal/edible plant-based products that seem to be safer to consume are ortiga (*Malvaviscus* sp.) [100], cilantro (*Coriandrum sativum*) [101], chope (*Vochysia* sp.) [100], and camu camu (*Myrciaria dubia*) [102] leaves; yarina (*Attalea* sp.) [100], cabbage (*Brassica oleracea*) [103], and radish (*Raphanus sativus*) [104] roots; and olive (*Olea europaea*) [90], mango (*Mangifera indica*) [56], and banana (*Musa paradisiaca*) [105] fruits.

However, the above examples do not represent a strict rule because what seems to be safer in one region can be contaminated in other places within the country; similarly, some products might not contain a specific metal, but they could also show high concentrations of other pollutants. For example, corn grains have been reported with very low (0.01 mg/kg) As concentrations in Arequipa [64], but with violations of the WHO standards for Pb in Lima [85]. Further investigations should evaluate each product, each metal, and each geographical location to develop a list of safe and non-safe plant products to consume based on those variables. Equally important, since most edible/medicinal plant-based products (and metals) included in this review are not regulated by the WHO standards, it is urgently necessary to regulate them to ensure that people in Peru are not at risk of ingesting unhealthy amounts of metals (see Section 2.1 for more details).

Figure 4 left illustrates the nationwide statistical distribution of metal concentrations, while Appendix A shows the mean, sample size, and standard deviation values of all plant-related metals found in this national review, indicating that Cd (37% of total analyses), Pb (28%), As (10%), Hg (7%), Cr (7%), and Ni (3%) are the metals most evaluated nationwide, with the rest of the contaminants representing 1% or less each. In terms of concentrations, Al, As, Ba, Cd, Cr, Ni, and Pb are the contaminants present with the highest values (national averages greater than 1 mg/kg) among plant-based food/medicinal products found in this analysis despite their health consequences for humans (e.g., [106]). Of particular interest is Al, a metal not regulated by the WHO (Appendix A), with a national average of 11.4 mg/kg. Moreover, it is important to keep in mind that only As, Cd, Sn, and Pb are regulated under the WHO standards for plants products, and even though some metals found in Peruvian plants are reported in lower concentrations (less than 1 mg/kg), they still might be harmful to humans. For example, one of the lowest concentrations was found for Se (national mean of 0.59 mg/kg) in glasswort (*Sarcocornia fruticosa*) leaves (an edible Peruvian plant that is not normed under the WHO’s standards) in Lima [97] (see Appendix A), but excessive ingestion of this metal can cause digestion issues [107].

Geographically speaking (Figure 4 (right)), the regions with more publications were Junin (18 studies, 11.5%) and La Libertad (16 studies, 10%), followed by Lima, Arequipa, Huánaco, and Ucayali. Only two departments (Lambayeque and Apurimac) showed a complete absence of scientific publications on this topic. According to the Ministry of Agriculture [108], the regions (or departments) with the largest agricultural surfaces are San Martín, Cajamarca, and Puno, suggesting that based on this analysis and for the latter region, plant products might be safer, the rivers used for irrigation purposes might be less polluted, or more likely, there has been no initiative to study bioaccumulation in those areas of the country. Thus, additional research is needed to determine the reasons for this lower number of publications in regions with high agricultural productivity. Taking into account the above, there is no clear correlation between high- or low-reported cases and location, at least from the perspective of administrative boundaries, so further studies should focus on the specific characteristics of local rivers used by downstream agricultural activities. For example, Tejada-Meza et al. [109] recently documented high tannery-derived metal contents in surface waters draining directly into the Chili River (Arequipa Region), whose waters are used to irrigate thousands of hectares of agricultural crops. Additionally, mining and other human activities could affect metal concentrations in river waters used to irrigate agricultural fields in this arid region of Peru (e.g., [110]), not mentioning that rivers in this part of the Andes naturally contain metals (e.g., [111]); for example, Regis et al. [112] reported high As concentrations in the Tambo river (Arequipa Region), and such metal has been reported in many food/medicinal products in that same region (Appendix A). Whatever the origin of the metal pollution is, the safest and most appropriate way to deal with this growing problem (metal bioaccumulation in plant-based food/medicinal products) is to treat polluted river waters before they are used for irrigation purposes, as suggested by Thomas et al. [113] to decrease Cd concentrations in Peruvian cacao beans. Furthermore, constructed wetlands (alone or in combination with other treatment methods such as membrane filtration [112]) might represent a cost-effective method for improving water quality in Peru [114]. Finally, stronger regulations are highly needed to demand proper treatment of agricultural crops and plants that Peruvians consume for food or medicinal purposes.

#### 2.3.2. Fish

Following plants, fish was one of the categories with the most reported bioaccumulation cases in Peru (a total of 284 documented analyses). Forty-seven species of fish were tested for bioaccumulation of metals in the country. Based on the literature found, fish were categorized under muscular tissue (202 analyses, 71%) and other parts of the body that Peruvians consume (82 analyses, 29%), with the latter including metal bioaccumulation in liver (12 analyses, 6%), kidney (2 analyses, 1%), gill (2 analyses, 1%), marrow (1 analysis, 0.5%), scale (1 analysis, 0.5%), bone (1 analysis, 0.5%), and mix and other parts including brain, gonad, and skin (6 analyses, 3%) (see Appendix A). This suggests that bioaccumulation in fish is well studied in Peru for muscular tissues (the most commonly eaten part of the animal). However, only 47 analyses (17%) were below the WHO’s MPLs, while 28 analyses (10%) were above the WHO standards. Most importantly, 209 analyses (74%) were not regulated under the WHO standards (see Section 2.2 for more details), suggesting an urgent need to create a stricter regulatory system for Peruvian fish. Moreover, most (61%) of the analyses were performed in river (continental) species, while the rest corresponded to ocean fish. In both cases, most analyses resulted to be not regulated (73 and 85% for river and ocean fish, respectively), while results below MPLs were documented occasionally (7 and 3%, respectively), with the reminding violating the WHO’s standards.

Among the most common fish-related food products containing metal concentrations above the WHO standards are carachama muscle meat (*Chaetostoma* sp.), with a record of 5.09 mg/kg of Pb in Huánaco [115], although various Peruvian authors reported Pb contents above the WHO standards in muscular tissue of rainbow trout (*Oncorhynchus mykiss*), as well as other parts of fish bodies (see Appendix A), since this metal is reported to bioaccumulate in trout elsewhere, although in lower concentrations, such as 3 mg/kg documented by McEneff et al. [116] in the United Kingdom. Furthermore, the highest Pb concentrations were found in the kidney of rainbow trout (12.54 mg/kg) [117], much higher than the 2.4 mg/kg documented in Norway [118]. Peruvians do consume fish kidneys for edible or medicinal purposes, suggesting high exposure to Pb. A comparison of metal concentration according to tissue type identified higher Cd in guts, kidney, and liver relative to muscle tissue and higher Hg in brain and liver tissue relative to muscle, liver, and kidney. While statistically relevant, these comparisons were from a small number of studies that may be impacted by confounding variables unaccounted for in the comparison.

Furthermore, the safest fish to eat is a function of the metal under subject as well as the geographical area where the species was caught; for example, muscle tissue from black rainbow trout has been reported to have Cd (1.52 mg/kg, non-regulated) in Apurímac [119], but none was found in the same species caught in Cajamarca [77]. Similarly, the native fish jumilla (*Parodon buckleyi*) has shown low regulated metal contents (Appendix A), suggesting that this species could be safer to consume; however, Rosales et al. [120] reported Cd concentrations higher than 1.6 mg/kg in the same species in Huánco. Brown trout (*Salmo trutta*) seems to have low metal content (Appendix A), inconsistent with other studies that have found high metal concentrations in the same species in Norway (e.g., [121]). Despite the above, perhaps the most important metal studied in fish is Hg. Previous studies documented 10 mg Hg/kg in brain and liver of Chilean jack mackerel (*Trachurus murphyi*) [122], and 1.2 and 1.1 mg Hg/kg of this toxic metal in muscular tissue from eastern pacific bonito (*Sarda chiliensis*) [123] and vulture catfish (*Calophysus macropterus*) [124], respectively. Mercury bioaccumulation in fish has been associated with illegal gold mining [125], a common practice in Peru that has been linked to Hg pollution in Peruvian rivers (e.g., [126]). In addition to Hg in fish, Ramos [127] reported a concerning 13.5 mg/kg of As in canned tuna (*Thunnus* sp.) sold in Arequipa, much higher than the maximum of 1.42 mg/kg reported by Andayesh et al. [128] in Iran. This As concentration in canned tuna represents a statistical outlier relative to other species in the data set. In conclusion, further studies are needed to confirm how safe the documented species are to ingest by humans, depending on the type of pollutant, type of fish, and region of origin.

Figure 5 (left) illustrates the nationwide statistical distribution of metal concentrations, while Appendix A shows the mean of metal concentrations found in fish in this synthesis, with Hg featured most prominently (28% of total analyses) followed by Pb (26%), Cd, (20%), and As (7%). However, there is also an isolated case documenting extremely high concentrations of Al (24.5 mg/kg) and Cd (13.5 mg/kg) concentrations in canned tuna (*Thunnus* sp.) [127] (as previously mentioned), which are much higher than what other authors have found in the Mediterranean Sea [129] and Brazilian markets [130], respectively, for the same species. Moreover, the metals that were reported at the highest concentrations nationwide were Ba (15.57 mg/kg) in rainbow trout [77], Cd in royal pleco (*Panaque nigrolineatus*) (1.62 mg/kg) [115] and jumilla (*Parodon buckleyi*) [120], and Pb in rainbow trout (10.45 mg/kg) [77]. It is important to keep in mind that only CH_3_-Hg, Sn (canned fish), and Pb (fish and four fish species) are regulated under the international WHO standards (see Appendix A and Section 2.1), indicating an urgent need to regulate other metals that have been found in high concentrations in Peru, to ensure that people in the country are not at risk of ingesting unhealthy amounts of these contaminants. Also, even though some metals found in Peruvian fish are reported in lower concentrations (less than 1 mg/kg), they still might be harmful to humans, so further studies should carefully evaluate each metal to define healthy MPLs.

In terms of habitat, metals considered for both river and ocean species were Pb (27 and 15% of total analyses performed in river and ocean fish species, respectively), Hg (26 and 59%), Cd (23 and 13%), and As (11 and 10%), and occasionally Ni (2%), Se (2%), and Cr (2%) in river species, and Al (3%) in ocean fish. Also, only river species were tested for CH_3_-Hg (8%) even though shark and tuna were studied and have published advisory levels by the WHO (see Appendix A). As illustrated in Appendix A, Hg, As, and Cd seem to be found at higher concentrations within ocean fish species, while Pb was documented with higher concentrations in river species. This difference should be further studied to determine what the sources of pollution are in both types of habitats to better understand why certain metals are detected and where.

Geographically speaking (Figure 5 right), the regions with more studies were Madre de Dios (8 studies, 15%), Loreto, Ucayali (7 studies each, 13% each), and Puno (5 studies, 9%), i.e., the eastern portion of the Andes where rainforests are located (i.e., freshwater fish species). A reasonable explanation for this is the fact that mining operations exist in those regions, leading to more metal concerns in surface waters, in addition to the naturally occurring metals that exist in rivers that flow from the Peruvian Andes, as previously discussed (see Section 2.3.1). Though in smaller quantities compared to eastern regions, Lima, Arequipa, Moquegua, and Tacna represent the majority of the national coastal fish-related research (Figure 5 right). Furthermore, additional studies are needed to determine, for example, the link between river water quality and metal content in fish located in both coastal and inland areas.

#### 2.3.3. Macroinvertebrates and Mollusks

Bioaccumulation in Peruvian macroinvertebrates and mollusks has been less documented than plants and fish, with a total of 26 studies and 214 analyses, of which 77 analyses (36%) corresponded to macroinvertebrates, and 137 (64%) were conducted in mollusks. Most analyses (62%) were performed in muscular tissue, while the remaining (38%) were done in other organs represented by digestive gland, exoskeleton, gill, gonad, and intestine (1 analysis each, 2% of total each), guts (4 analyses, 7%), gutted body (11 analyses, 18%), hepatopancreas and quelas (2 analyses each, 3% each), and whole body (3 analyses, 5%). However, only 20 species were evaluated (5 macroinvertebrates and 15 mollusks), suggesting that further research is needed to investigate metal bioaccumulation in other species from this food category.

Appendix A shows the metal concentrations of all macroinvertebrates and mollusks and their body parts, including products below (24 analyses, 11%) and above (5 analyses, 2%) the WHO standards and those currently not regulated (185 analyses, 86%) (see Section 2.1 for more details). Following the WHO’s standards, no macroinvertebrate species evaluated is currently regulated even though high metal contents were reported such as 75.7 and 37.3 mg/kg of As in muscular tissue from Chilean crab (*Hepatus chilensis*) and Cancer crab (*Romaleon polyodon*), respectively, in Ica [131]; 280 mg/kg of Al in muscular tissue from Changallo shrimp (*Cryphiops caementarius*) in Arequipa [132]; and 12.4 mg/kg of Pb in pincers of hairy Crab (*Cancer setosus*) in Ancash [133], just to name a few concerning examples. Moreover, even though some Peruvian mollusks are regulated for Cd and only two analyses showed concentrations above MPLs (2.7 and 2.1 mg/kg in muscular tissue from magellan mussel (*Aulacomya ater*) in Arequipa and Lima, respectively [134]), the majority of metals detected in mollusks are not regulated by the WHO. Among the most common food products related to this classification containing metal concentrations below international standards are Chilean scallop (*Argopecten purpuratus*) and navajuela (*Tagelus dombeii*) [135]. Moreover, Figure 6 (left) illustrates the nationwide statistical distribution of metal concentrations, while Appendix A shows the mean of all macroinvertebrate- and mollusk-related metals considered in this review, with Pb (28%), Cd (26%), Hg (12%), and Ni (7%) being the contaminants that have been most documented for this food category in Peru. The metals that were reported in the highest concentrations nationwide (above 10 mg/kg) were Al (mean of 112.9 mg/kg), As (15.0 mg/kg), and Ba (27.7 mg/kg), while Cd, Sn, Pb, Se, and Ti were documented with averages above 1 mg/kg (see Appendix A). When evaluating the entire macroinvertebrate and mollusk list according to metal, Al represents an outlier. The Al value is attributable to one species, changallo shrimp (*Cryphiops caementarius*), which was evaluated in two different studies in Arequipa [132,136]. However, it is important to keep in mind that only Cd and Sn (canned food) are listed under the international WHO standards (Appendix A), and even though some metals found in in this category are reported in lower concentrations (less than 1 mg/kg, Figure 6 left), they still might be harmful to humans.

As expected, scientific publications were distributed among most coastal departments (Figure 6 right), led by Moquegua and Tacna, though research has been developed in only seven regions nationwide. No studies were found in the coastal departments La Libertad and Lambayeque, suggesting that more research is needed to evaluate the macroinvertebrates and mollusks that exist in these regions. In fact, La Libertad is known for having high metal contents in rivers flowing into the Pacific Ocean (e.g., [137]).

#### 2.3.4. Mammals

Bioaccumulation in Peruvian mammals has been less documented than other food categories, with a total of only two species (see further down), 13 scientific publications, and 105 analyses. Both species and their products (e.g., milk) are consumed by humans, where 66 analyses (63%) corresponded to body parts from Alpaca (*Vicugna pacos*), a very common protein source in Peru. Body part analyses were found for muscular tissue (14 analyses, 13% of total); liver, lung, and bone (12 analyses each, 11% each); and fiber (4 analyses, 4%). The remaining 39 analyses (37%) were conducted on cow’s (*Bos taurus*) milk (35 analyses, 33% of total analyses) and cheese (4 analyses, 4%). However, even though “alpaca meat” is not included in the WHO’s standards, national authors documented the presence of four metals (Hg, Cd, Pb, and one analysis showing As) in this Peruvian food type, highlighting the findings by Orellana et al. [138] (English literature), who documented high Cr concentrations in Huancavelica (see Appendix A). Moreover, cuy (*Cavia porcellus*, another common food source in Peru) is not regulated either, and no metal analysis was found for it. In addition to cuy, no scientific publications were found for cow and pig meat, both of which are a common food source for Peruvians, suggesting an urgent need to evaluate metal bioaccumulation in these national protein sources. The need is also clear for regulating such animals and metals found in this review to ensure that people in Peru are not at risk of ingesting unhealthy amounts of contaminants.

All this suggests that bioaccumulation in mammals is not well studied in the country; similarly, it is important to know how these animals are affected by the ingestion of contaminated plant-based feed or contaminated water. In fact, though not being part of this review, Peruvian authors have reported metal contents (Hg, Al, As, Ba, Cd, Cr, Ni, and Pb) in plant species commonly used for forage (e.g., [76,92,139,140,141,142,143,144,145,146,147,148,149,150]) that could contribute to bioaccumulation in edible mammals. Considering this, further studies should focus on the specific characteristics of local rivers used for animal drinking, metal concentrations in soils, and forage-type plants that farmers use to feed domestic mammals or regions where range-free mammals naturally exist.

Appendix A shows the metal concentrations of all mammals, their body parts, and processed foods, including products below (2 analyses, 2%) and above (13 analyses, 12%) the WHO standards and those currently not regulated (90 analyses, 86%) (see Section 2.2 for more details). Among the most common mammal-related food products containing metal concentrations above international standards is milk. Nationwide, Pb was above the norm for most analyses conducted on milk, agreeing with the findings by Ismail et al. [151], who developed an international review (excluding Latin American countries) and found evidence of concerning metal concentrations in milk from Egypt, Iran, Serbia, Poland, Hungary, and Croatia. Based on the findings by those authors, the highest global Pb concentration was documented in Egypt (7.02 mg/kg, by Malhat et al. [152]), while Peruvian records were 12.76 and 11.61 mg/kg as reported in Puno by Barcena [153] and Paredes [147], respectively, representing a concern that needs to be further studied and addressed in the country. Based on the mammal data set, there were no statistical differences between species, materials tested, or analytes.

Figure 7 left illustrates the nationwide statistical distribution of metal concentrations, while Appendix A shows the mean of all mammal-related metals considered in this review, with Pb (30%), Cd (29%), Hg (21%), and As (20%) as the only contaminants that have been documented in Peruvian mammals. All metal concentrations presented nationwide averages were less than 1 mg/kg. However, it is important to keep in mind that only Pb is regulated under the international WHO standards for mammals (Appendix A), and even though some metals found in Peruvian mammals are reported in lower concentrations (less than 1 mg/kg, Figure 7 left), they still might be harmful to humans (for example, the WHO’s MPLs for Pb is 0.1 mg/kg).

Based on Figure 7 (right), the regions with more scientific publications were Puno and Lima, followed by six more departments with only one study, while the rest of the country did not develop any research in the topic. However, grazing is indeed characteristic of more regions within the entire national territory [154], this being another indication that metal bioaccumulation in this food category is not well studied in Peru.

#### 2.3.5. Other Food Categories

With a total of 16 species and 93 analyses, other categories not discussed above included (a) bees, honey, pollen, and propolis (5 analyses, 5%); (b) eggs (25 analyses, 27%); (c) linchen and fungi (46 analyses, 49%); (d) Algae (12 analyses, 13%); and (e) the edible frog *Batrachophrynus macrostomus* (4 analyses, 4%) (Appendix A). This category represents very common food/medicinal products in Peru, with Pb (4.3 mg/kg) contents reported by Párraga et al. [155] in propolis (a Peruvian cold medicine), a medicinal product not regulated under the WHO standards. To put things into perspective, González-Martín et al. [156] reported Pb concentrations of around 0.1 mg/kg in propolis from Spain (more than 40 times smaller than what has been reported in Peru).

Similarly, non-regulated eggs from chicken (*Gallus domesticus*) and Japanese quail (*Coturnix japonica*) showed the presence of Pb (4 mg/kg), Cd (0.15 mg/kg), As (0.04 mg/kg), and Hg (0.5 mg/kg) [157,158,159,160]. To put Peruvian results into perspective, Aliu et al. [161] reported 0.07, 0.006, and 0.001 mg/kg of Pb, Cd, and Hg, respectively, in Kosovo, while Kirov et al. [162] documented maximum Pb and Cd concentrations of 0.27 and 0.05 mg/kg, respectively, in Bulgaria, suggesting that metal contents in Peruvian eggs are significantly higher than concentrations reported elsewhere.

Moreover, though some are regulated according to the WHO standards, medicinal/edible lichen and fungi also contained Pb (19 mg/kg), Ni (4.9 mg/kg), Cr (20.1 mg/kg), Cd (0.82 mg/kg), As (4.8 mg/kg), and Al (92.7 mg/kg). Even though no studies were found regarding metal bioaccumulation in edible/medicinal lichen, Protano et al. [163] reported maximum Pb, Ni, Cr, Cd, and As concentrations of 28.4, 8.4, 8.5, 0.43, and 1.51 mg/kg, respectively, in transplanted Lichen (*Pseudovernia furfuracea*) at sites adjacent to a solid-waste landfill in central Italy, indicating that Cr, Cd, and As concentrations reported in Peru are superior to those documented in highly contaminated sites.

The above studies suggest that bioaccumulation in those food/medicinal categories is well studied in Peru, although there is a clear need to evaluate metal contents in the muscular tissue (meat) of chicken and other birds commonly eaten in the country and how they are affected by ingesting contaminated feed. In fact, corn (an important feed type for domestic chicken) has been reported to have high metal contents in Peru (see Section 2.3.1).

Appendix A shows the metal concentrations of all other food/medicinal categories, including products below (4 analyses, 4%) and above (12 analyses, 13%) the WHO standards, and those currently not regulated (77 analyses, 83%) (see Section 2.1 for more details), with Pb and Cd being above the norm for most analyses done in lichen and fungi.

Figure 8 left illustrates the nationwide statistical distribution of metal concentrations, while Appendix A shows the mean of all metals considered in this review under other food categories, with Pb (29%), Cd (25%), Cr (12%), Ni (13%), Hg (11%), As (8%), Al (2%), and Se (1%) being the contaminants that have been documented in Peru. Clearly, Al (nationwide mean of 67.7 mg/kg), Cr (6.1 mg/kg), Ni (2.4 mg/kg), and Pb (5.5 mg/kg) represent a high risk for consumers, as they all present high national averages. Similarly, statistical differences include lower Cd in eggs relative to other categories and more Pb in lichen and fungi relative to the other categories. Further statistical comparisons were unreliable due to the small data set size.

However, it is important to keep in mind that only Pb is regulated under international WHO standards for this food category (Appendix A), and even though some metals found in Peru are reported in lower concentrations (less than 1 mg/kg, Figure 8 left), they still might be harmful to humans. Moreover, it is concerning that one of the most popular food types in Peru—chicken-based food products—have only been studied in a handful of regions (see Figure 8 (right)), suggesting that in addition to evaluating metal content in bird meat, further studies should investigate metal contents in other regions of the country.

## 3. Conclusions

This extensive literature search provides for the first time a complete database of metal bioaccumulation in food/medicinal products that are normally consumed in Peru, representing the starting point for further research on this important topic. Similarly, the study provides solid fundaments to create a better regulatory system that can lead to safer diet for locals, also ensuring the avoidance of future exporting penalties from Peruvian producers in the international markets.

This review represents an important contribution to the English literature on what has been reported in Spanish in terms of metal presence and bioaccumulation in Peruvian food/medicinal products. An extensive literature search from 231 scientific publications shows that Peruvian food/medicinal products normally consumed in the country can have high concentrations of metals due to bioaccumulation, which could result in health issues. In fact, this review reveals that metal concentrations reported in Peru are in many cases much higher than what has been reported elsewhere. Even though a small fraction of analyses showed metal concentration below MPLs, the biggest concern is the fact that most metals and products being consumed are not included in the WHO’s advisory lists, suggesting the urgent need for stronger regulations to ensure the safety of locals in Peru.

Some regions within the country and some food products are notably a problem that needs to be addressed. There is a clear lack of investigations in terms of spatial distribution, metals involved, and species evaluated, with popular protein sources such as meat from cuy, chicken, and pork as good examples. Similarly, though certain common metals are highly present, other metals that are also toxic and are not included in any international standard system need to be regulated. Out of 1907 documented analyses and 239 species investigated, the results show that the majority (63%) of nationwide metal bioaccumulation has been evaluated in plant-based food/medicinal products followed by fish (15%), macroinvertebrates and mollusks (11%), mammals (6%), and other categories (5%).

### Recommendations

Despite the clear inconsistency among the international standard systems (in terms of metals considered and food categorization), the results suggest that the country needs to use the information from these other regulatory and advisory bodies to tailor and specify country-specific regulations that protect the health and well-being of its citizens and to consider routes of exposure and local food stock (e.g., alpaca) that might not be captured in other regions.

The results also indicate the need to replicate this study for metal bioaccumulation in other sources, such as animal feed, to understand how those affect metal concentration in animals that people consume at a national level. Similarly, more research should focus on how river water quality and anthropic activities can affect metal bioaccumulation in locally grown food products as well as consider the findings of the Peruvian private educational sector.

## Figures and Tables

**Figure 1 foods-13-00762-f001:**
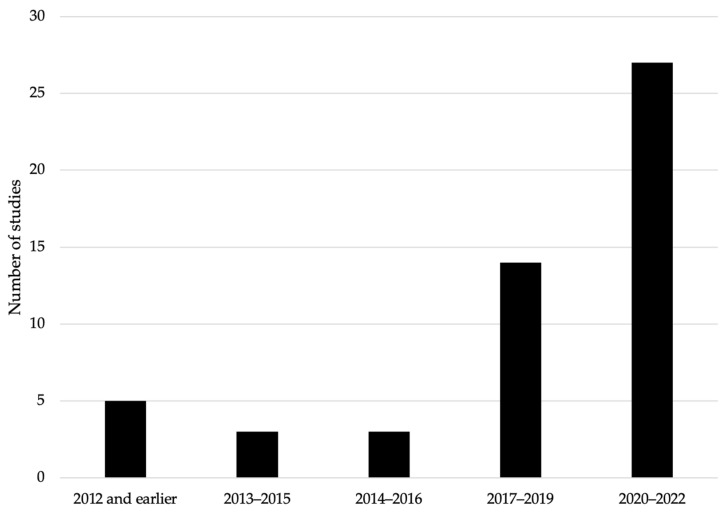
Temporal increases in peer-reviewed publications in English journals (up to 2022) that focus on the presence of metals and bioaccumulation in consumer products in Peru.

**Figure 2 foods-13-00762-f002:**
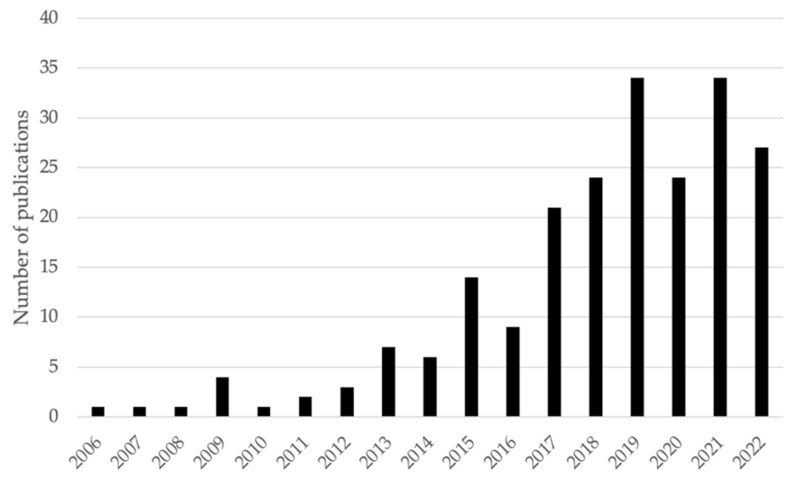
Temporal evolution of metal bioaccumulation-related scientific publications (in Spanish) in Peruvian food/medicinal products.

**Figure 3 foods-13-00762-f003:**
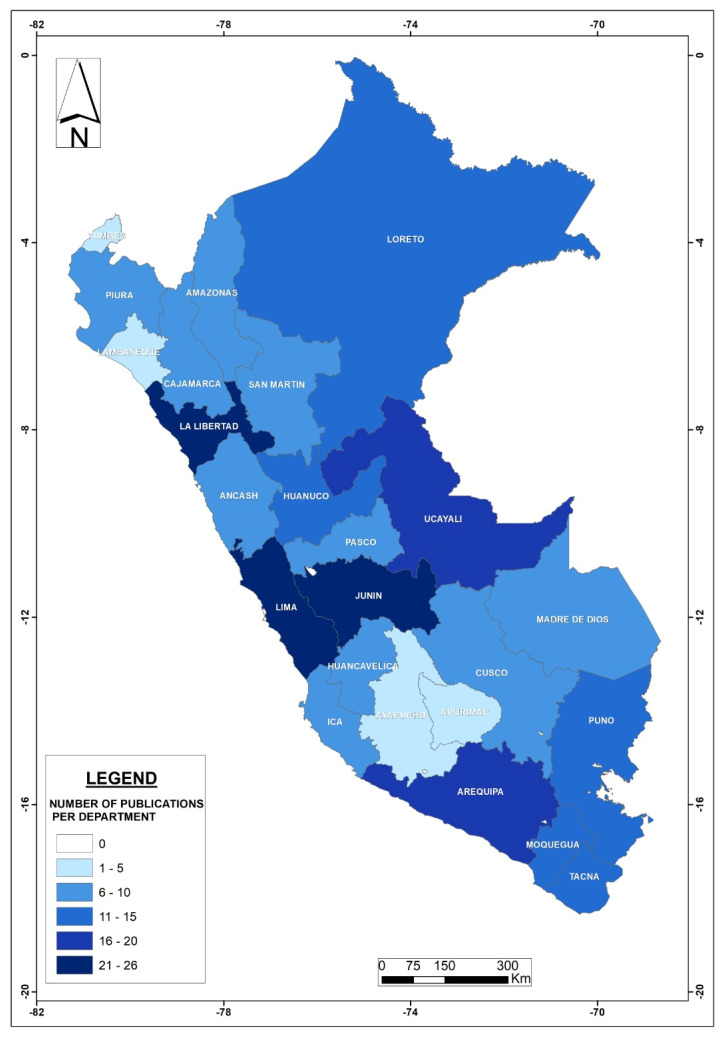
Geographic distribution of bioaccumulation-related studies in Peru per administrative department.

**Figure 4 foods-13-00762-f004:**
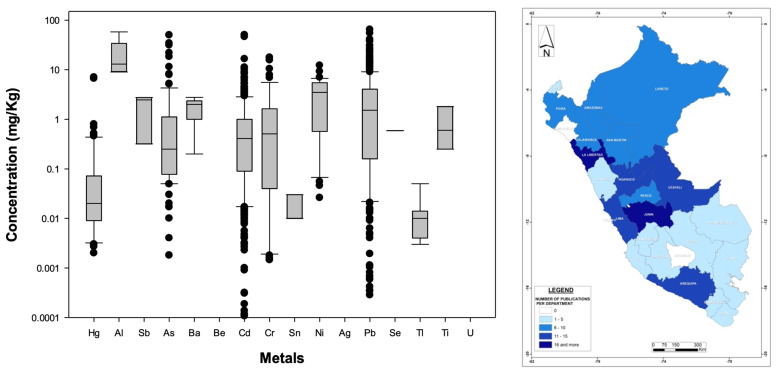
(**Left**): Distribution (boxplots) of nation-wide metal concentrations documented in Peruvian plant-derived food/medicinal products. (**Right**): Geographic distribution of published bioaccumulation-related studies developed for plant-based food/medicinal products in Peruvian administrative departments (please refer to Figure 3 for details). The WHO standards are not displayed because they are specific to the food/medicinal type.

**Figure 5 foods-13-00762-f005:**
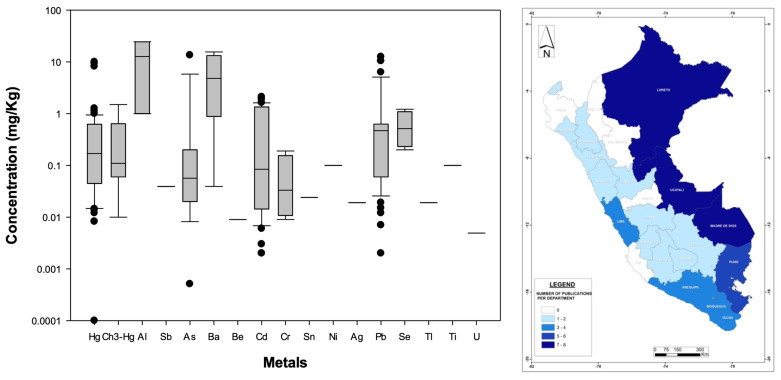
(**Left**): Distribution (boxplots) of nationwide metal concentrations documented in Peruvian fish-derived food/medicinal products. (**Right**): Geographic distribution of published studies related to bioaccumulation developed for fish-based food products in Peruvian administrative departments (please refer to Figure 3 for details). The WHO standards are not displayed because they are specific to the food/medicinal type.

**Figure 6 foods-13-00762-f006:**
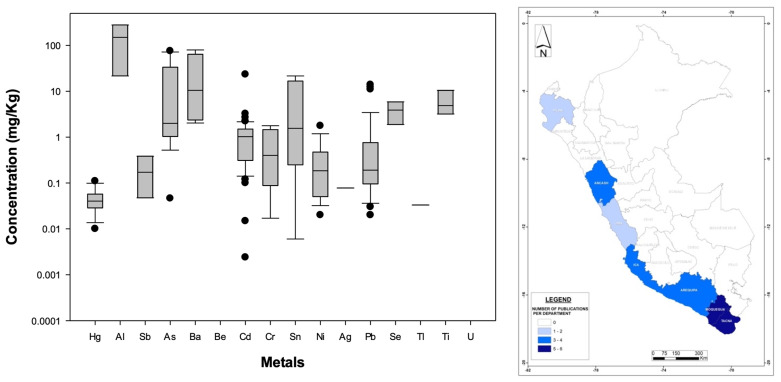
(**Left**): Distribution (boxplots) of nationwide metal concentrations documented in Peruvian macroinvertebrates and mollusks. (**Right**): Geographic distribution of published bioaccumulation-related studies developed for macroinvertebrate- and mollusk-based food products in Peruvian administrative departments (please refer to Figure 3 for details). The WHO standards are not displayed because they are specific to the food/medicinal type.

**Figure 7 foods-13-00762-f007:**
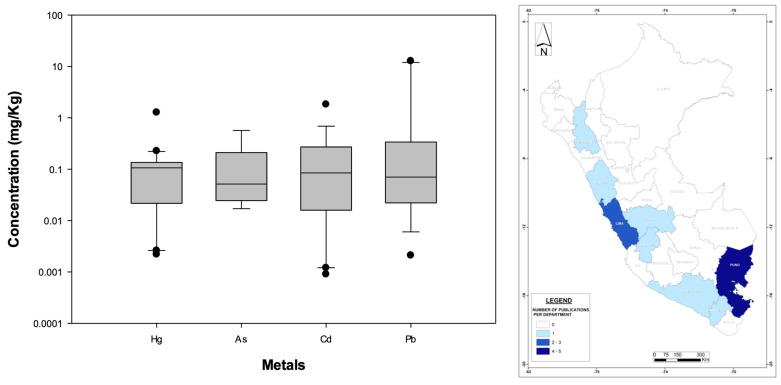
(**Left**): Distribution (boxplots) of the nationwide metal concentrations documented in Peruvian mammals. (**Right**): Geographic distribution of published bioaccumulation-related studies developed for macroinvertebrate- and mollusk-based food products in Peruvian administrative departments (please refer to Figure 3 for details). The WHO standards are not displayed because they are specific to the food/medicinal type.

**Figure 8 foods-13-00762-f008:**
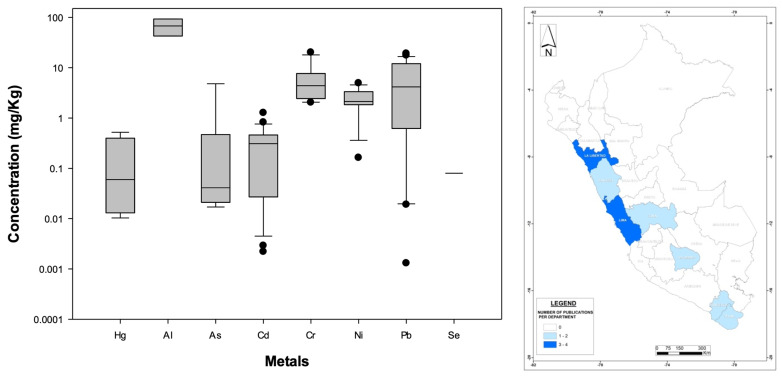
(**Left**): Distribution (boxplots) of nationwide metal concentrations documented in other food categories in Peru. (**Right**): Geographic distribution of published bioaccumulation-related studies developed for food products based on other categories in Peruvian administrative departments (please refer to Figure 3 for details). The WHO standards are not displayed because they are specific to the food/medicinal type.

**Table 2 foods-13-00762-t002:** Percentage of analyses below the WHO’s MPLs (black), above the MPLs (red), and not regulated (blue) for all food/medicinal products found in this Peruvian review.

	Hg	CH_3_-Hg	Al	Sb	As	Ba	Be	Cd	Cr	Sn	Ni	Ag	Pb	Se	Tl	Ti	U
Plants	100	-	100	100	97	100	100	41	100	100	100	100	32	100	100	100	100
	0	-	0	0	1	0	0	33	0	0	0	0	14	0	0	0	0
	0	-	0	0	2	0	0	27	0	0	0	0	54	0	0	0	0
Fish	100	100	100	100	100	100	100	100	100	100	100	100	0	100	100	100	100
	0	0	0	0	0	0	0	0	0	0	0	0	37	0	0	0	0
	0	0	0	0	0	0	0	0	0	0	0	0	63	0	0	0	0
Macroinvertebrates and mollusks	100	-	100	100	100	100	100	48	100	100	100	100	100	100	100	100	100
	0	-	0	0	0	0	0	9	0	0	0	0	0	0	0	0	0
	0	-	0	0	0	0	0	43	0	0	0	0	0	0	0	0	0
Mammals	100	-	-	-	100	-	-	100	-	-	-	-	53	-	-	-	-
	0	-	-	-	0	-	-	0	-	-	-	-	41	-	-	-	-
	0	-	-	-	0	-	-	0	-	-	-	-	6	-	-	-	-
Others	100	-	100	-	100	-	-	70	100	-	100	-	67	100	-	-	-
	0	-	0	-	0	-	-	26	0	-	0	-	22	0	-	-	-
	0	-	0	-	0	-	-	4	0	-	0	-	11	0	-	-	-

**Table 3 foods-13-00762-t003:** Metals (and number of food categories that include each of them) considered in different international standards compared to those found in this national analysis. As(1) = Total As; As(2) = Inorganic As.

	Hg	CH_3_-Hg	Al	Sb	As(1)	As(2)	Ba	Be	Cd	Cr	Sn	Ni	Ag	Pb	Se	Tl	Ti	U
WHO	2	4			6				17		8			41				
Cuba	8	2			46				47		17			60				
Mercosur	5				52				47		2			53				
Australia–NZ	13				2	2			5		1			6				
S. Africa	2	2		1	4	2			14		3			58				
EU	34					4			58		5			62				
Total	64	8	0	1	110	8	0	0	188	0	36	0	0	280	0	0	0	0

**Table 4 foods-13-00762-t004:** MPLs (mg/kg) for some metals under the “Fish” and “Fruits” categories in different international food standards.

		Cuban	WHO	Australian–NZ	EU	Brazilian	S. African	Sweden
Fish	Hg	1.00	-	1.50	0.50	0.50	-	0.20
	As	1.00	-	2.00	-	1.00	-	-
	Cd	0.10	-	-	0.05	1.00	-	0.10
	Pb	0.30	-	0.50	0.30	2.00	0.30	0.50
Fruit	Hg	0.00	-	-	-	-	-	-
	As	0.30	-	-	-	-	-	-
	Cd	0.05	-	-	-	-	-	0.05
	Pb	0.10	0.10	0.10	0.10	0.50	0.10	0.50

**Table 5 foods-13-00762-t005:** Distribution of different bioaccumulation-related publication types in Peru for all food/medicinal categories evaluated.

Publication Category	Plants	Fish	Macroinvertebrates and Mollusks	Mammals	Others	Total
Undergraduate theses	95	18	7	13	10	143
Master’s theses	21	5	4	4	2	36
PhD dissertations	12	2	0	3	2	19
Peer-reviewed journals	13	6	2	0	1	22
Reports	3	5	3	0	0	11
Total	144	36	16	20	15	231

## Data Availability

No new data were created or analyzed in this study. Data sharing is not applicable to this article.

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
