# Peer review of "Heavy Metal Bioaccumulation in Peruvian Food and Medicinal Products"

_foods, 2024, doi:10.3390/foods13050762_

Round 1
Reviewer 1 Report
Comments and Suggestions for Authors
Dear authors, I have analyzed the manuscript “Heavy Metal Bioaccumulation in Peruvian Food and 2 Medicinal Products” and I would like to make the following observations:
1. It is an interesting study with a large number of bibliographic indications analyzed but which needs a restructuring, because it is very difficult to follow and no clear conclusions can be drawn.
2. I recommend redoing the abstract in a more synthetic form in which to clearly specify the purpose, the innovative character and the relevance of the research. It is also necessary to present one quantifiable result, relevant to the entire research. It has too many characters for an abstract...
3. 3. The introduction is rather chaotic; it has no logical thread. I recommend the creation of a synthetic table in which it is established for each metal studied, the source of origin and the harmful effects, but a bibliographic indication.
4. A section in which all abbreviations are defined from the beginning is recommended.
5. I definitely recommend the use of special statistical analysis software with which to interpret information. For a journal of this caliber, you can't just use excel...
6. It would be necessary at the beginning of the study to have some sort of table of contents for an easier follow-up of the information
7. I strongly recommend going through the information for authors of the journal to fix the way the citations are entered in the text and for the bibliography section. Accuracy is needed for the entire manuscript.
8. I strongly recommend going through the information for authors of the journal to fix the way the citations are entered in the text and for the bibliography section. Accuracy is needed for the entire manuscript.
9. Line 38 "covered 239 species", reformulate it is not clear
10. Line 47 "though commonly non-regulated" reformulate, explain why and give a bibliographic reference.
11. Lines 54-58 transfer them as the last paragraph in the introduction section.
12. Line 83 "As national and export food demands increased in Peru", reformulate. The amount of metals increased not because of the demand, but because of more checks. Likewise in figure 1.
13. I don't understand what you wanted to demonstrate with table 1, revise. I recommend giving the content another form. It's too scholastic. A statistics program is needed.
14. The information presented in figures 2-9 will have to be reconsidered for a statistical interpretation.
15. The conclusions section must have only this role. If you also want recommendations, create a separate section. The conclusions must be reformulated to be clear, concise and representative for the research.
16. Considering the mentioned, I think that the manuscript needs a major revision.
Author Response
Reviewer 1
The authors deeply thank Reviewer 1 for taking the time and effort to improve our manuscript. We have addressed each comment as follow and we hope that Reviewer 1 feels like this improved version is good enough for publication, as we also had to consider the requests and opinion from other reviewers.
- It is an interesting study with a large number of bibliographic indications analyzed but which needs a restructuring, because it is very difficult to follow and no clear conclusions can be drawn. Response: The authors thank the reviewer for this comment. It is hard to see where the author is asking for improvement, but we have modified the results, discussion, and conclusions to improve the clarity of the research. We hope that the reviewer will be satisfied with this improved version.
- I recommend redoing the abstract in a more synthetic form in which to clearly specify the purpose, the innovative character and the relevance of the research. It is also necessary to present one quantifiable result, relevant to the entire research. It has too many characters for an abstract... Response: The authors thank the reviewer for the suggestion. After carefully reading the Abstract, we noticed that it contains what was requested by the reviewer. However, we have modified the abstract to address the request and we hope that this new version satisfied the reviewer-
- The introduction is rather chaotic; it has no logical thread. I recommend the creation of a synthetic table in which it is established for each metal studied, the source of origin and the harmful effects, but a bibliographic indication. Response: The authors thank the reviewer for this suggestion, which has been addressed accordingly.
- A section in which all abbreviations are defined from the beginning is recommended. Response: The authors thank the reviewer and agree. A list of abbreviations was added.
- I definitely recommend the use of special statistical analysis software with which to interpret information. For a journal of this caliber, you can't just use excel... Response: The authors thank the reviewer for this suggestion and agree. Statistical analysis software was applied to the data, with the respective interpretation.
- It would be necessary at the beginning of the study to have some sort of table of contents for an easier follow-up of the information. Response: The authors thank the reviewer for the suggestion and agree. A table of contents was incorporated.
- I strongly recommend going through the information for authors of the journal to fix the way the citations are entered in the text and for the bibliography section. Accuracy is needed for the entire manuscript. Response: The authors thank the reviewer for the recommendation. Citations have been entered according to the formatting guidelines of Foods MDPI. This process will be finalized during proofreading stages.
- Line 38 "covered 239 species", reformulate it is not clear. Response: The authors thank the reviewer and agree. The sentence has been modified for clarity.
- Line 47 "though commonly non-regulated" reformulate, explain why and give a bibliographic reference. Response: The authors thank the reviewer for this comment. We are referring to the WHO advisory lists corresponding to the food categories included in this review, which lack regulations for those metals. We have modified the sentence for clarity.
- Lines 54-58 transfer them as the last paragraph in the introduction section. Response: The authors thank the reviewer and agree. The request was addressed accordingly.
- Line 83 "As national and export food demands increased in Peru", reformulate. The amount of metals increased not because of the demand, but because of more checks. Likewise in figure 1. Response: For Line 83: The authors thank the reviewer for this comment. However, there was a misunderstanding, as in this sentence we are talking exclusively about the evolution of food demands (not metals). To clarify, we added a reference. For Figure 1: Similarly, in this figure we are showing how English-written metal bioaccumulation peer-reviewed publications have increased during the last decade or so (not metals themselves). We don’t see the need to address this comment, though we are happy to modify the text if the reviewer believes is needed.
- I don't understand what you wanted to demonstrate with table 1, revise. I recommend giving the content another form. It's too scholastic. A statistics program is needed. Response: The authors thank the reviewer for this comment. As discussed in the first sentence, Table 1 is intended to show the reader the proportion of analyses found to be above and below WHO standards, as well as those not included in the WHO standards (i.e. not regulated in Peru), pointing out that the majority of the metals found in food/medicinal products normally consumed in Peru aren’t regulated, so the creation of new regulations is urgent. We discussed how to address this suggestion and concluded that, since there are so many categories and metals to show, a figure or chart would be even more confusing for the reader. However, we added more discussion in the text to add clarity to the reader.
- The information presented in figures 2-9 will have to be reconsidered for a statistical interpretation. Response: The authors thank the reviewer for this suggestion and agree. Though the authors believe that the current figures are important for the reader to understand what metals could represent a safety problem, we have incorporated new figures (boxplots) to provide an idea of data distribution at a national level, while the original figures were included in new SI files. Those changes were incorporated toward the document, accordingly.
- The conclusions section must have only this role. If you also want recommendations, create a separate section. The conclusions must be reformulated to be clear, concise and representative for the research. Response: The authors thank the reviewer and agree. We have modified the conclusions and included recommendations in a separate section, as suggested.
Reviewer 2 Report
Comments and Suggestions for Authors
Why did the authors consider the thesis?
- Line 66: Please, put a space between the number and unit.
- Line 67: I suggest to specify the term “high contents”. How much higher was the cadmium level than legally established?
- Lines 67-68: The same of before.
- Line 71: Again, I suggest to underline and specify the contents.
- Line 73: “Masco et al. (YEAR)”.
- Line 76: Does this statement consider studies that statistically bind mercury to cancer? If so, how many studies do they say so? On what level of significance?
Lines 77-82: The same of before.
Line 90: This statement is quite important and strong. Only one reference states this?
Line 96: I suggest to replace “Though” with “Although”.
Line 109: Galagarza et al. (YEAR).
Line 111: I suggest to put “Inorganic pollutants”.
Line 112: “13 studies”, where are the references?
Line 138: I suggest to replace “Dangerously” with “Relevantly” or something similar. If the concentrations are statistically higher than the values established by WHO standards, I suggest to use the term “significantly”.
Line 151: MPL stands for?
Table 2: I suggest to put the units in the didascalia of the table. It will be easier for the reader to follow the story.
Author Response
Reviewer 2
The authors deeply thank Reviewer 2 for taking the time and effort to improve our manuscript. We have addressed each comment as follow and we hope that Reviewer 2 feels like this improved version is good enough for publication, as we also had to consider the requests and opinion from other reviewers.
Why did the authors consider the thesis? Response: The authors thank the reviewer for this comment. As stated in the Introduction (and Section 2.2), undergraduate and graduate theses represent the majority of research results at a national level, which should not be ignored. Herein, we have compiled such information from 50 national universities.
- Line 66: Please, put a space between the number and unit. Response: The authors thank the reviewer for the suggestion, which has been addressed.
- Line 67: I suggest to specify the term “high contents”. How much higher was the cadmium level than legally established? Response: The authors thank the reviewer for the suggestion. We have incorporated the legal concentration limit (0.004 mg/kg)
- Lines 67-68: The same of before. Response: The authors thank the reviewer for the suggestion. Values above MPLs were incorporated for different food products banned from the European Union for having too much metals.
- Line 71: Again, I suggest to underline and specify the contents. Response: The authors thank the reviewer and agree. The text was modified accordingly.
- Line 73: “Masco et al. (YEAR)”. Response: According to the formatting rules of Foods MDPI, year must be replaced by the number of the citation, in this case. The authors thank the reviewer for understanding.
- Line 76: Does this statement consider studies that statistically bind mercury to cancer? If so, how many studies do they say so? On what level of significance? Response: The authors thank the reviewer for this improvement. We have added the review by Skanly et al. (2022), who concluded that mercury can indeed be linked to cancer. The authors hope that the reviewer thinks that this is enough proof, as talking about this in too much details would get out of the scope of this manuscript.
Lines 77-82: The same of before. Response: The authors thank the reviewer for the comment and agree. The text was modified accordingly, including also a review by Suhair et al. (2019) on the effects of lead on human health, as well as other studies.
Line 90: This statement is quite important and strong. Only one reference states this? Response: The authors thank the reviewer for this comment and agree. Though there are some studies focusing on the topic, we decided to simply erase the sentence.
Line 96: I suggest to replace “Though” with “Although”. Response: The authors thank the reviewer and agree. The change has been addressed accordingly.
Line 109: Galagarza et al. (YEAR). Response: According to the formatting rules of Foods MDPI, year must be replaced by the number of the citation, in this case. The authors thank the reviewer for understanding.
Line 111: I suggest to put “Inorganic pollutants”. Response: The authors thank the reviewer and agree. The change has been addressed accordingly.
Line 112: “13 studies”, where are the references? Response: The authors thank the reviewer for this comment. Reviewing the document again, there were actually 16 studies evaluated in the review done by Galagarza et al. (2021), and they were all incorporated in the citation list accordingly, as requested. The authors apologize for the error.
Line 138: I suggest to replace “Dangerously” with “Relevantly” or something similar. If the concentrations are statistically higher than the values established by WHO standards, I suggest to use the term “significantly”. Response: The authors thank the reviewer for the suggestion, which has been addressed.
Line 151: MPL stands for? Response: MPL stands for Maximum Permissible Limits, which was already defined in Line 109.
Table 2: I suggest to put the units in the didascalia of the table. It will be easier for the reader to follow the story. Response: The authors than the reviewer for the comment. There are no units, as each value in the table represents the number of of food categories that contain each metal. This is already explained in the didascalies.
Reviewer 3 Report
Comments and Suggestions for Authors
- While the article is generally well-written, certain sections could benefit from more concise wording and a clearer exposition of key points.
- A more detailed exploration of the potential health impacts of the various metals found in food products would strengthen the article's relevance to public health concerns.
- The article could benefit from a clearer distinction between the findings and their implications, ensuring that the results are not overshadowed by the discussion.
- Adding visual elements, such as graphs or charts, to represent key data findings might enhance the reader's understanding and engagement.
Author Response
Reviewer 3
The authors deeply thank Reviewer 3 for taking the time and effort to improve our manuscript. We have addressed each comment as follow and we hope that Reviewer 3 feels like this improved version is good enough for publication, as we also had to consider the requests and opinion from other reviewers
While the article is generally well-written, certain sections could benefit from more concise wording and a clearer exposition of key points. Response: The authors thank the reviewer for the suggestion. Even though the manuscript was carefully reviewed by native English-speaking professors, we have run a grammar software and additional changes were incorporated.
A more detailed exploration of the potential health impacts of the various metals found in food products would strengthen the article's relevance to public health concerns. Response: The authors thank the reviewer for this suggestion and agree. To make things easier for the reader (and following the request from another reviewer), we have incorporated a table with the health effects of each metal.
The article could benefit from a clearer distinction between the findings and their implications, ensuring that the results are not overshadowed by the discussion. Response: The authors thank the reviewer for this suggestion. Discussion was improved toward the document.
Adding visual elements, such as graphs or charts, to represent key data findings might enhance the reader's understanding and engagements. Response: The authors thank the reviewer for the suggestion and agree. Figures were improved to provide a better idea to the reader about of the distribution of the data.
Round 2
Reviewer 1 Report
Comments and Suggestions for Authors
Accept in present form!